# Optimizing hepatitis B diagnosis for mothers in a low-resource setting: A field pilot of Xpert point-of-care viral load testing in Ugandan antenatal clinics

Linda Kisaakye Nabitaka[1]*, Viola Kasone[2•], Emmanuel Olal[3•], Judith Kyokushaba[2], Philip Kasibante[1], Susan Nambozo[2], Aidarayaan Namakula[2], Susan Nabadda Ndidde[2], Ponsiano Ocama[4], Isaac Ssewanyana[2]

**1** STD/AIDS Control Program, Ministry of Health, Kampala, Uganda, **2** National Health Laboratory and Diagnostic Services, Ministry of Health, Kampala, Uganda, **3** Infectious Diseases Cluster, Clinton Health Access Initiative, Kampala, Uganda, **4** Makerere University College of Health Sciences, Kampala, Uganda

☺ These authors contributed equally to this work.
* lkisaakye@gmail.com

## Abstract

Mother-to-child transmission accounts for most new hepatitis B (HBV) infections in high-burden regions globally. The World Health Organization recommends antiviral prophylaxis for pregnant women with high hepatitis B viral load levels, but access to viral load testing remains limited and untimely in many countries, due to centralized testing. We assessed the operational feasibility, accessibility, and cost of implementing the Xpert HBV DNA assay in antenatal care settings. This was a two-phase study at 10 high-volume hospitals across nine regions of Uganda. Phase I verified the diagnostic accuracy of Xpert HBV DNA assay against the COBAS Taqman platform, while Phase II assessed feasibility, usability, and cost of integrating PoC testing into routine antenatal care using GeneXpert systems. Quantitative data were extracted from facility registers, entered into ODK, and analysed using SPSS 19 and Excel. Qualitative data, consisting of feedback from study teams, were analysed thematically. Xpert HBV DNA assay demonstrated full concordance with the COBAS platform. Overall, 96.7% of samples were processed, 92% returned to providers, and 61% of mothers received their results on the same day. Of 181 pregnant women provided HBV DNA testing, 12% had viral loads >200,000 IU/mL, thus eligible for antiviral prophylaxis. Among pregnant women eligible for prophylaxis, 14 had received antiviral prophylaxis. Birth records were available for 15 pregnant women, and 73.3% of their newborns had received the hepatitis B birth dose vaccine. Coordination between ANC and laboratory services, including the timely dispatch of results to pregnant women, was generally smooth and enabled prompt decision-making. However, some challenges were reported, such as stockout of screening kits and competing testing priorities within laboratories. At USD 15.37 per test, its cost was comparable to centralized

**Data availability statement:** Minimal data set that contains all raw data required to replicate study results has been included as part of the Supporting documents.

**Funding:** This study was supported by Cepheid and the Ministry of Health, Uganda. Cepheid provided GeneXpert cartridges, technical support for platform malfunctions, and a research grant (no grant number assigned) to SI a through a fiduciary agency to support field study teams. The Ministry of Health, Uganda provided in-kind support for all testing-related infrastructure and consumables, including GeneXpert platforms, and phlebotomy supplies (vacutainers and syringes), in addition to hepatitis B screening commodities and technical personnel who led the study. The funders had no role in study design, data collection and analysis, decision to publish, or preparation of the manuscript.

**Competing interests:** The authors have declared that no competing interests exist.

testing (USD 15.26), but it offered advantages such as reduced delays and fewer client visits. The Xpert platform offers a timely, accurate, operationally feasible solution for accessing hepatitis B antiviral prophylaxis or treatment, improving quality of care for hepatitis B-positive mothers. With rapid turnaround time and cost comparable to centralized testing, it presents a valuable tool for improving HBV prevention and treatment in Uganda and similar resource-limited settings.

## Introduction

Daily, over 3000 people contract hepatitis B globally [1], a figure comparable to the HIV incidence [2]. Over the past 15 years, the World Health Organization (WHO) has issued a series of frameworks, global strategies, guidelines, and position papers to elevate the prevention of mother-to-child transmission (PMTCT) of hepatitis B (HBV) as a public health priority [3–7]. These documents provide normative guidance, strategic targets, and milestones to accelerate the elimination of viral hepatitis B as a global health threat by 2030 [5].

Mother-to-child transmission (MTCT) remains the primary driver of new HBV infections, particularly in high-prevalence regions such as the WHO Africa and Pacific regions [1]. However, the PMTCT toolkit has expanded since the release of the 2009 WHO position paper recommending the hepatitis B birth dose vaccine [3]. In addition to a timely hepatitis B birth dose and routine infant vaccination, the WHO recommends antiviral prophylaxis during pregnancy for women with high hepatitis B viral load, typically starting at 28 weeks of gestation or as early as possible, and continuing until three months postpartum [7].

HBV DNA quantification is the preferred method for assessing eligibility for treatment and prophylaxis during pregnancy [7,8]. In many countries, viral load testing is typically conducted on centralized high-throughput platforms [7], which, while accurate and efficient, are often located far from primary health care facilities, where antenatal care is delivered. This contributes to delays in the return of test results and subsequent initiation of prophylaxis [9,10]. Rapid turnaround of HBV viral load results is essential to ensure that pregnant women eligible for prophylaxis receive it by 28 weeks of gestation or earlier. Nevertheless, persistent diagnostic access barriers underscore the need for decentralized testing approaches to expand coverage and reduce delays [6,11].

Uganda bears one of the highest HBV burdens globally, with a national prevalence of about 4.3% [12]. At the country level, in 2019, the Ministry of Health adopted a triple elimination strategy targeting the vertical transmission of HIV, syphilis, and HBV through integrated antenatal care (ANC) [13]. Since the strategy's launch, programmatic data show that HBV screening at ANC has risen steadily from 12% in 2020 to over 56% in 2023. However, fewer than 40% of HBsAg-positive pregnant women receive viral load testing, limiting opportunities for timely identification and prophylaxis.

Currently, Uganda leverages its HIV sample referral network for HBV DNA testing through a hub and spoke model, wherein peripheral health facilities (spokes)

send samples to the Central Public Health Laboratory (CPHL) in Kampala, the country's capital city [14,15]. Although this model offers processing efficiency, it is associated with significant delays. A recent study by Tahir et al (2023) reported a median Turn Around Time of 48 days (IQR 29–73) from sample collection to receipt of results hampering timely clinical decision-making [16]. Anecdotal reports suggest that in some cases, results are returned only after pregnant women have delivered, thereby missing a critical window for maternal prophylaxis.

Point-of-care (PoC) viral load testing offers a promising solution to close this gap. Among the seven commercially available HBV viral load assays globally, only two are designed for PoC use: Xpert HBV Viral Load Care (Cepheid Inc.) and Truenat HBV (Molbio Diagnostics) [17]. The Xpert platform, a CE-marked rapid molecular assay, is the only PoC HBV DNA platform currently available in Uganda. Uganda extensively employs the GeneXpert platform for tuberculosis, HIV, HPV, and COVID-19 diagnostics [18], offering an opportunity to leverage existing infrastructure for HBV DNA testing.

The GeneXpert platform has demonstrated high diagnostic accuracy for HBV DNA, with a limit of quantification as low as 10IU/mL [19,20]. However, evidence of its real-world feasibility in antenatal settings for the prevention of mother-to-child transmission is limited. This study assessed the operational performance, accessibility, feasibility, and cost of using the Xpert HBV Viral Load assay to strengthen the PMTCT of HBV in Uganda, leveraging the country's existing GeneXpert network. As Uganda prepares to scale HBV PMTCT services, real-world evidence on PoC solutions is critical to inform policy and investment decisions.

## Methods

### Study design and settings

This was a two-phase, pragmatic, cross-sectional, mixed-methods implementation study conducted between April and December 2024 across ten high-volume public and private not-for-profit hospitals in Uganda. Participating sites included the central laboratory (CPHL), where Phase I was conducted at the laboratory responsible for testing all viral load samples from public health facilities. This aimed to verify the diagnostic accuracy of the Xpert HBV Viral Load (VL) assay against the reference COBAS AmpliPrep/COBAS TaqMan platform, as required by the Ministry of Health, Uganda, before field implementation.

Phase II was implemented across 10 purposively selected hospitals based on high antenatal care (ANC) volume, documented location in regions with a high burden of hepatitis B, and the presence of substantial GeneXpert platform capacity (16 modules) to minimize disruption to other molecular tests such as those for TB, human papillomavirus, and COVID-19 testing. For Phase II, the aim was to evaluate the feasibility, usability, acceptability, and cost of incorporating the assay into routine ANC workflows using Uganda's decentralized GeneXpert network.

### Study population and sampling

Phase I clinical performance evaluation involved a specimen panel of 50 archived samples selected using a simple random sampling approach, including HBV-positive plasma specimens, as described in 1 below. These samples were obtained from the National Biorepository at CPHL, stored at -80°C for 3 weeks, and had sufficient volume per the protocol. The manufacturer recommends that archived plasma specimens remain stable when frozen at -80°C to 20°C for 6 weeks. The sample size for analysis was reduced from the originally planned 248–50. This decision complies with CPHL IVD evaluation guidelines, which set testing requirements for this type of assay. Likewise, for this CE-marked test, this sample size is considered sufficient to ensure a robust performance evaluation (Table 1).

Phase II rolled out to the 10 sites: enrolled HBsAg-positive pregnant women aged 18 years and above attending ANC services at study sites. A purposive target of 20 participants per site (~200 total) was determined based on the availability of test cartridges provided by Cepheid Inc. In addition, healthcare workers involved in ANC and laboratory services were purposively selected to participate in qualitative interviews and focus group discussions (FGDs).

**Table 1. Specimens for the clinical evaluation.**

| Type of specimens | Number of specimens |
|---|---|
| **Specimens from HBV-infected individuals** | |
| < 20 000 IU/mL, including | 15 |
| *<2 000 IU/mL* | *15* |
| *2000-20 000* | *0* |
| > 20 000 IU/mL, including | 35 |
| *20 000–200 000 IU/mL* | *17* |

## Study procedures

**Phase I: Diagnostic accuracy evaluation.** The diagnostic accuracy of the GeneXpert platform has been extensively validated in prior studies. Therefore, this phase of the study aimed only to verify its performance within the local context. We assessed agreement with the platform currently used for routine viral load testing, using 50 archived samples. The 50 archived plasma samples were retrieved from the CPHL biorepository and tested using the Xpert HBV VL assay. The samples had been archived between 24th February 2024 and 12th March 2024. Tests were done on 12th March-14th March 2024.

HBV-positive specimens were characterized using the COBAS AmpliPrep-COBAS TaqMan Hepatitis B Virus Quantitative test (Roche Diagnostics GmbH) on plasma. The result obtained with this test served as the reference; no additional testing was necessary. Results were compared with those previously obtained from the COBAS platform. Viral load results were log-transformed to evaluate agreement and clinical performance. The test was performed according to IFU Reference GXHBV-VL-CE-10. Testing was conducted by two operators who demonstrated competency in using the assay before starting the evaluation. Clinical specimens described above were tested individually on the assay under evaluation using one lot.

**Phase II: Participant identification and enrollment.** At the ten study sites, pregnant women attending ANC were screened for HBsAg. Participants were recruited from 29th April to 5th December 2024. Eligibility for enrolment included: a positive hepatitis B surface antigen test, gestational age above 24 weeks, and providing informed consent to participate in the study. Blood samples were collected and processed for HBV DNA testing on-site using the Xpert platform. Participant information was recorded using Open Data Kit (ODK) pretested electronic questionnaires.

**Sample collection and onsite testing.** Each eligible participant provided 5 mL of venous blood collected at ANC by the midwife in BD Vacutainers and Plasma Preparation Tubes. The midwife sent samples to the laboratory for on-site testing using the Xpert HBV VL assay. Samples were tracked using unique identifiers on viral load request forms to ensure results were linked back to the right pregnant women. Testing was integrated into routine clinical workflows, and ANC services continued per national guidelines.

**Operational monitoring and implementation feedback.** Throughout the study, operational indicators such as uptake, test completion rates, and turnaround time were monitored by a data manager. Monthly virtual feedback meetings were conducted between the study and health facility teams. After the study implementation, a virtual focus group discussion (FGD) was held with 21 participants, including site-level staff and national program officers. A semi-structured engagement guide was used to explore implementation domains, including workflow integration, test usability, turnaround time, data handling, and perceived patient experience.

**Data quality and verification.** To enhance data completeness, the study team conducted a follow-up visit at facilities with incomplete records. The visit helped to retrieve or verify information on participant identifiers, test results, and any data fields that were not captured during routine electronic data collection.

**Visual summary of testing models.** To contextualize the intervention, **Fig 1** compares sample flow under the conventional hub-based testing model versus the integrated point-of-care approach. The left panel depicts the standard multi-tier sample transport pathway to the Central Public Health Laboratory (CPHL). The right panel shows same-day, onsite testing using GeneXpert platforms, enabling faster clinical decisions and timely linkage to care.

**Follow-up of mother-infant pairs for prophylaxis and hepatitis B birth dose vaccination.** Pregnant women eligible for prophylaxis or treatment were later followed up by the study coordinator to ascertain whether they had been successfully linked to prophylaxis or chronic treatment, and whether their newborns had received hepatitis B birth dose vaccination, in line with Uganda guidelines.

### Data analysis

**Pha se I.** The viral load values (copies/ml) were transformed to log10 copies/ml. Descriptive statistics were presented as mean values ± standard deviations (SDs). The Deming regression method was used to compare the two methods. A Bland–Altman plot was used to visualize the mean differences between the quantification assays [21]. Furthermore, the sensitivity and specificity of the Xpert HBV Viral Load were assessed against the COBAS CAPCTM/Taqman as the gold standard at a clinical threshold >20,000 IU/ML.

**Phase II.** Quantitative data were analyzed using SPSS version 26 and Excel, and summarized as means and standard deviations for continuous variables, and as percentages and median plus interquartile ranges for categorical variables, using tables and graphs. Thematic analysis was used to analyze qualitative data describing the technical and operational feasibility, acceptability, and the cost of testing per patient.

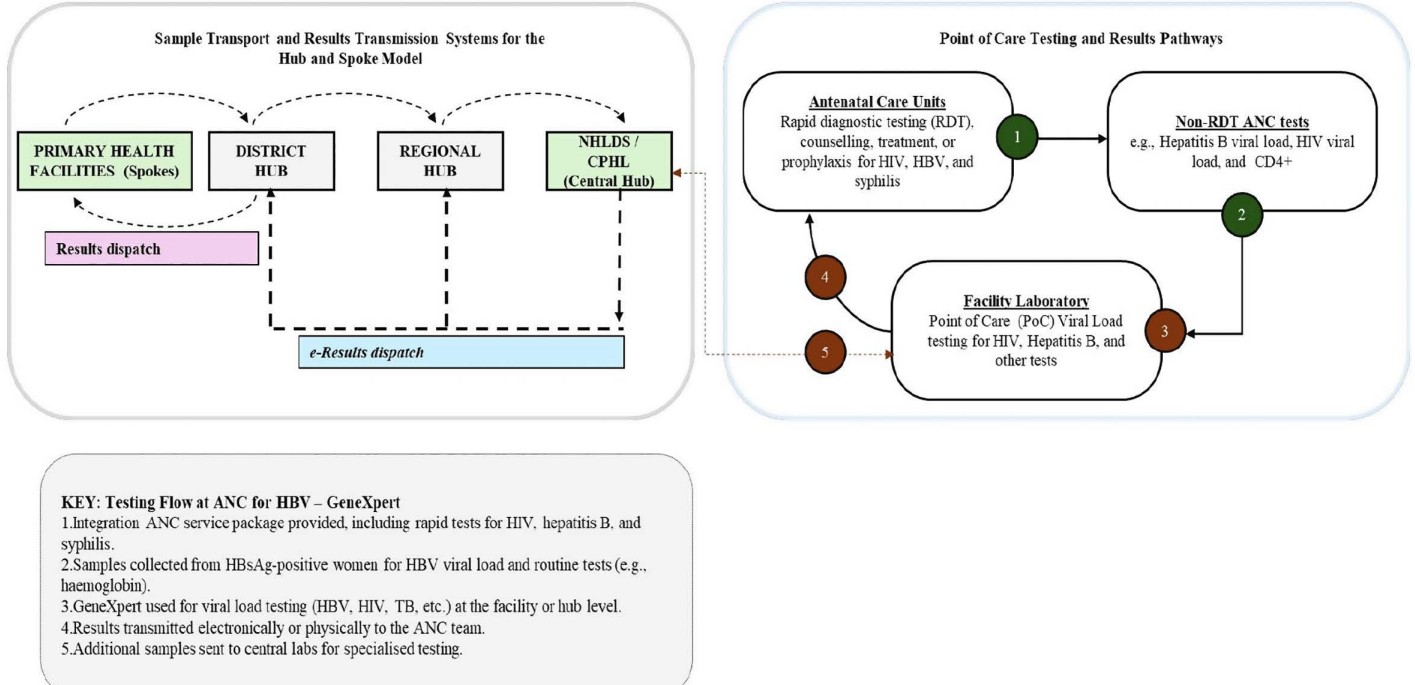

**Fig 1.** *Comparing conventional sample flow with PoC GeneXpert flow.*

**Global Public Health**

## Ethical consideration

All participants provided written informed consent for study participation. The study was approved by the Uganda Ministry of Health, the Uganda National Health Laboratories Research Ethics Committee (UNHL-2023–73), and the Uganda National Council for Science and Technology (HS4687ES). Archived samples used in Phase I were anonymized before testing, while all the clients who participated in Phase II of the study provided written informed consent. Only participants 18 years and above were included in this study. Data was stored securely with coded identifiers to ensure participant confidentiality.

## Results

### Phase I

The level of agreement and correlation with the reference COBAS AmpliPrep-COBAS TaqMan Hepatitis B Virus Quantitative test was within acceptable limits, with a correlation coefficient of 0.99, as shown in Table 2. The Xpert HBV Viral Load on Gene Xpert systems demonstrated good performance and successfully met all the requirements for use as a test for quantification of HBV Viral load in patients. We did not measure turnaround time in Phase I, as both platforms were located at the CPHL and had comparable sample processing times. See S1 File for further details.

### Phase II

**Sociodemographic and clinical characteristics of participants.** A total of 259 pregnant women were screened for potential enrolment into the study. Sociodemographic characteristics and other relevant background health information of the pregnant women were collected to ensure eligibility and understand the implementation of other areas of triple elimination. Of the 259 women, 13 were ineligible: 1 tested negative for hepatitis B, 3 were already known HBV-positive and on treatment, and 7 were less than 24 weeks' gestation; two had HIV HBV co-infection and were already on a TDF-containing regimen. Among the 246 eligible participants, 188 had complete clinical assessment reports and 181 provided blood samples for hepatitis B viral load testing; thus, they had laboratory assessment reports, as shown in Table 3. However, only 123 participants had complete clinical evaluation and corresponding laboratory results, enabling matched data analysis.

Notably, 100% of pregnant women were screened for HIV and syphilis, in line with Uganda's triple elimination strategy for HIV, HBV, and syphilis. Most (96.8%) expressed a willingness to initiate HBV treatment if eligible, demonstrating high treatment readiness. Despite this, only 14.4% (27 out of 188) of the mothers knew their HBV status before the current pregnancy. Vaccination coverage was notably low: only 15.4% (29 out of 188) reported having ever received the HBV vaccine, 134 (71.3%) had never been vaccinated, and the rest did not know or had missing data on vaccination status. All women younger than 24 years should have ideally benefited from the national pentavalent vaccine rolled out in 2002, yet only 7% in this age group had been vaccinated, see S2 File, for further details on Phase II findings.

**Table 2. Agreement Between GeneXpert and Reference Assay for HBV Viral Load Quantification.**

| | | Results of the comparator assay | | |
| --- | --- | --- | --- | --- |
| | | Positive (> 20,000 IU/mL) | Negative (< 20,000 IU/mL) | |
| **Results of Xpert HBV Viral load** | > 20,000 IU/mL | 35 | 0 | 35 |
| | < 20, 000 IU/mL | 0 | **15** | 15 |
| | | 35 | 15 | 50 |

• Positive percent agreement: 35/35 = 100% (95% CI: 90%-100%).

• Negative percent agreement: 15/15 = 100% (95% CI: 78.2%-100%).

**Table 3. HBV Testing, Awareness, and Treatment Acceptance.**

| Variables | | Frequency (N = 188) | % |
|---|---|---|---|
| Hepatitis B viral load sample collected at this visit | No | 7 | 3.7 |
| | Yes | 181 | 96.3 |
| Mother is willing to start HBV treatment if eligible | No | 3 | 1.6 |
| | Yes | 182 | 96.8 |
| | Missing data | 3 | 1.6 |
| Knowledge of HBV status before this pregnancy | No | 161 | 85.6 |
| | Yes | 27 | 14.4 |
| History of HBV treatment | No | 186 | 98.9 |
| | Yes | 2 | 1.1 |
| HBV vaccinated | No | 134 | 71.3 |
| | Yes | 29 | 15.4 |
| | Don't know | 7 | 3.7 |
| | Missing data | 18 | 9.6 |
| Age of vaccinated pregnant women (N = 29) | 15-19 years | 1 | 3.4 |
| | 20-24 years | 8 | 27.8 |
| | 25-29 years | 7 | 24.1 |
| | 30-34 years | 11 | 37.9 |
| | 35-39 years | 2 | 6.9 |

**Turnaround time.** Specifically, 96.7% of samples were sent from ANC to the laboratory on the same day they were collected, 99% of samples sent to the laboratories were received and processed at the laboratory on the same day, and 92% were returned to the service provider at MCH on the same day. Of the 92% of results sent to the ANC service provider, 61% were delivered to the pregnant women on the same day as the sample collection date (Fig 2 and S3 File).

Linkage to prophylaxis and hepatitis B birth dose vaccination: Among the 188 pregnant women who completed clinical interviews, 96.3% provided blood samples for hepatitis B viral load testing. Of these, 18% had an undetectable viral load, 2% between 20,000 and 200,000, 12% greater than 200,000, while 69% had a viral load below 20,000. This means that only 12% of mothers were eligible for prophylaxis under Uganda's hepatitis B PMTCT guidelines. The 12% (22 women) eligible for prophylaxis were followed up by the study coordinator at the end of the study to assess their pregnancy status and newborn outcomes. Of these, no further facility records existed for four women. Of the remaining 18 women with available records, 14 (77.8%) received antiviral prophylaxis. However, three (21.4%) were lost to follow-up later. By the end of the study, 15 women, including those who had not started prophylaxis, had delivered. Of their newborns, 11 (73.3%) received the hepatitis B birth dose vaccination.

## Field-level experiences with GeneXpert as a point-of-care platform for HBV load testing

Overall findings from the field-level implementation of GeneXpert for hepatitis B viral load testing reveal a largely positive experience across participating health facilities in Uganda. Health workers (laboratory personnel and midwives) reported that the platform was easy to use, significantly improved turnaround times, and enhanced clinical decision-making at the point of care. The integration of testing into maternal services strengthened patient engagement and enabled the timely initiation of treatment. However, these successes were tempered by recurring challenges, including stockouts, data entry constraints, prioritization constraints with other tests, e.g., TB, HPV, and HIV, and gaps in patient education. These insights highlight both the potential and the practical limitations of decentralized HBV testing in real-world settings and offer valuable lessons for future scale-up efforts. Key quotes from these engagements are captured in Table 4.

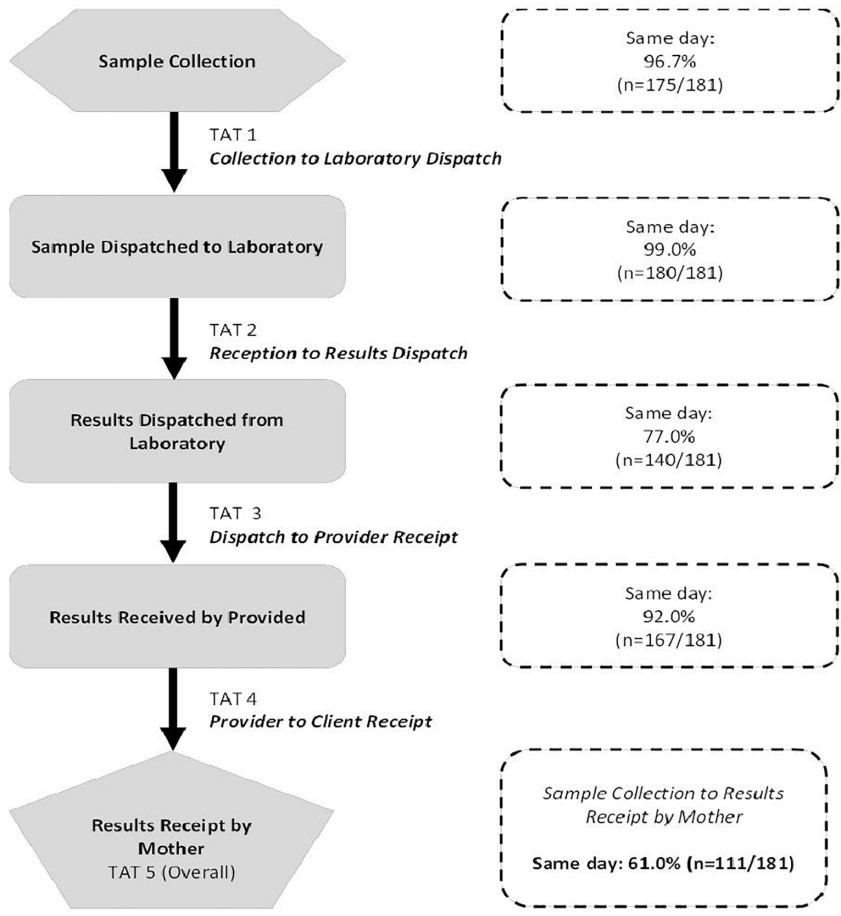

**Fig 2. Flow diagram showing different time points from sample collection to results receipt by pregnant women (mothers).**

## Ease of use and operational experience

Participants from several sites consistently described the GeneXpert platform as user-friendly, especially when standard operating procedures were followed. Although some participants reported initial challenges such as cartridge rejection or interpreting log-format results, these were recurring early-use issues that were quickly resolved with technical support. Overall, participants indicated rapid adaptation and the ability to perform the tests with minimal errors thereafter.

## Results turnaround time and clinical decision-making

GeneXpert significantly improved turnaround times for HBV viral load results, often delivering same-day results. This rapid feedback enabled timely clinical decision-making, particularly in initiating antiviral treatment. Integration of ANC and hepatitis clinics in some regions facilitated same-day testing, result receipt, and care.

## Patient care and clinical integration

The availability of on-site testing was reported to have greatly enhanced patient satisfaction and improved health worker confidence in HBV management. Some sites ran integrated HIV/HBV clinics, enhancing continuity of care. Mothers appreciated access to testing, counselling, and follow-up in a single visit.

**Table 4. Field-level experiences of using the GeneXpert PoC Platforms for hepatitis B viral load testing.**

| Theme | Key Quotes |
|---|---|
| *Ease of use and operational experience* | "The machine is straightforward to use when we were running those tests... most of the time we are within the turnaround time." Laboratory Personnel, <br> "For the case of Jinja, we did not have any challenges utilizing the point-of-care machine." Clinician |
| *Results turnaround time and clinical decision-making* | "Eligible pregnant women received the point-of-care test and the results in one to two hours." Midwife <br> "It helped in clinical judgment... because some patients may not have clear symptoms, but the viral load could be very high." Clinician |
| *Patient Care and Clinical Integration* | "It was a very good initiative as we could get results the same day, and the mothers would get their results that same day." Midwife <br> "This simplified hepatitis B care... If we had this point of care earlier on, by now we would have gone far." Clinician |
| *Result transmission and follow-up mechanisms* | "Through LabXpert... clinicians have been informed of the results... and patients also receive notifications." Laboratory personnel <br> "If you're free, you can go and pick up your results so that you can clear up your patient to go home." Clinician |
| *Training and Capacity Building Needs* | "The training wasn't really that sufficient… we would need further training if we could organize." Laboratory Personnel <br> "There were issues to do with the interpretation of results... we need the retraining." Central Coordination Team |
| *Prioritization of Samples on GeneXpert* | "If they found the machine was operating, they would wait… if it was free, they would run the sample." <br> "We have several tests going on... finding free modules was sometimes hard." Laboratory Personnel |
| *Stockouts and Supply Chain Challenges* | "We've had no testing kits for some time… now we have kits and have begun doing so." Clinical Lead <br> "We experienced stockouts of testing kits." Clinical Lead |
| *Data Entry and System Integration Issues* | "The Internet was disturbing... which caused them not to share data efficiently." <br> "Our major challenge, truthfully, was not capturing the data that was bringing the sample because of this issue." Laboratory Personnel |
| *Community Education and Patient Misconceptions* | "Some mothers believed they got hepatitis B through greeting or sharing food… some came depressed." Midwife <br> "We tell mothers we are testing the number of viruses in your blood… to help us decide how to care for you." Midwife |

## Result transmission and follow-up mechanisms

LabXpert's SMS notification system helped bridge communication gaps by alerting clinicians and patients when results were ready. However, this was not uniformly implemented across all sites, and reliance on manual result delivery persisted in facilities with weaker lab-clinic integration.

## Implementation challenges and system gaps

Despite strong performance and general acceptance of GeneXpert as a point-of-care platform for hepatitis B viral load testing, several cross-cutting implementation challenges were identified across participating health facilities. These included gaps in staff capacity to operate the GeneXpert, lack of sample prioritization protocols, frequent stockouts of HBsAg kits, data system limitations, and patient misconceptions. These factors impacted the consistency and quality of service delivery in some settings. The sections below highlight these key challenges and are illustrated with direct feedback from field teams.

### Training and capacity building needs

Training quality and sufficiency varied across sites. While some staff felt confident operating the GeneXpert platform, others expressed the need for refresher sessions, especially in interpreting viral load results. This occasionally led to delays or uncertainty in clinical decisions.

### Prioritization of samples on GeneXpert

In high-volume sites, sample prioritization emerged as a key challenge due to multiple disease programs competing for limited machine access. In the absence of standard operating procedures, staff made decisions based on urgency or availability, sometimes resulting in delays in processing hepatitis B samples.

### Stockouts and supply chain challenges

Several facilities reported frequent stockouts of HBsAg test kits, which disrupted testing continuity and delayed patient enrollment. In some cases, they had to suspend testing temporarily until new supplies were delivered.

### Data entry and system integration issues

Challenges with data entry were noted in multiple sites, largely due to poor internet connectivity, incompatible devices, or a lack of clarity on how to use the digital platforms. This affected real-time data capture and synchronization between clinical and laboratory information systems.

### Community education and patient misconceptions

Several mothers held misconceptions about hepatitis B, including beliefs about casual transmission through handshakes or shared meals. In some cases, this led to anxiety or stigma following diagnosis. Health workers highlighted the need for strengthened community education and patient counseling.

### Hepatitis B viral load testing: commodity & cost comparison

This study aimed to assess the itemized costs of delivering hepatitis B viral load testing at the point of care using the GeneXpert platform, which was being piloted. We used an **ingredient-based costing approach** to estimate the economic costs from the provider's perspective. This method involved identifying the total quantities of goods and services used in delivering the intervention and multiplying these quantities by their respective unit prices. The cost focused on direct inputs related to diagnostic commodities: test kits, reagents, consumables, and associated transportation logistics. The analysis excluded indirect costs such as personnel, patient-incurred costs, and overhead costs such as electricity, waste management, and water. The estimated unit cost per hepatitis B viral load test was USD 15.37 using the Cepheid GeneXpert platform, which is comparable to that of the COBAS platform currently being used in the country. See S4 File for an itemized cost comparison between the two platforms

## Discussion

Timely hepatitis B viral load testing and prompt transmission of results to clinical teams are crucial for the optimal initiation of antiviral prophylaxis for pregnant women who test positive for hepatitis B surface antigen (HBsAg) during antenatal care, aiming to prevent hepatitis B mother-to-child transmission (MTCT) [22]. We aimed to evaluate the operational feasibility and effectiveness of a point-of-care platform for HBV DNA testing across 10 high-volume district and regional referral hospitals in nine regions of Uganda. Our findings indicate that use of the GeneXpert PoC platform substantially reduced result turnaround time compared with the centralized testing model, which has been associated with prolonged delays, with a median turnaround time of 48 days (IQR 29–73) from sample collection to result receipt, hindering timely clinical

decision-making [16]. In this study, nearly all samples were received and processed, with results returned to providers on the same day, demonstrating the efficiency of PoC platforms in real-world settings. The PoC also enabled same-day results for over three-fifths of pregnant women (61%). While analytical processing times for GeneXpert and centralized COBAS platforms are broadly comparable, the decentralized placement of GeneXpert at the point of care or near point of care eliminates pre-analytical delays related to sample transportation and handling, thus resulting in a substantially shorter turnaround time for clinical decisions. We note, however, that 39% of samples were not returned to mothers on the same day. While the overall turnaround time remained substantially shorter than that reported in other studies for centralized testing, these delays were partly attributable to workload constraints and the need to prioritize other samples, including TB and HIV, in line with multiplexing policies at health facilities. It is also possible that some mothers left as soon as they received all the clinical and obstetric care, before the results were released and only returned on a subsequent day, thus the delay.

Timely results significantly reduce loss to follow-up and facilitate quick access to life-saving prophylaxis, thereby improving HBV prevention of mother-to-child transmission outcomes at scale. Specifically, we found that 14 of the 22 pregnant women eligible for prophylaxis received it. Among newborns with traceable facility records, 73.3% (11/15) received the hepatitis B birth dose vaccination to protect against hepatitis B infection. Our findings are consistent with a systematic review by Gu et al. (2024), which combined results from five countries (Egypt, Ethiopia, The Gambia, Sierra Leone, and Zambia). It found that HBV PoC DNA testing provided results in less than one day, significantly improving timely linkage to treatment initiation among eligible women [23].

We further evaluated the operational characteristics of the GeneXpert PoC platform according to the WHO Assured Criteria to determine if it is affordable, sensitive, specific, user-friendly, rapid, robust, equipment-free, and deliverable to end-users [24]. All participants widely reported that the GeneXpert platform was easy to operate, enabled timely decision-making, and supported rapid initiation of antiviral treatment. Moreover, its sensitivity and specificity were comparable to a conventional centralized testing platform. The operational ease of using the GeneXpert platform concurs with other studies from across Africa, including Kenya [25] and Zimbabwe [26], where it was shown that the platform required limited training to adopt and was easy to use. Similarly, a study from Nigeria found that the GeneXpert platform enhanced clinical decision-making in South Africa [27], demonstrating great accuracy and operational feasibility for use in clinical settings [28].

Considering that all ten sites had 16 module Xpert platforms, a part of Uganda's network of about 295 GeneXpert platforms used for tuberculosis, HIV, early infant diagnosis, HPV, and COVID-19 testing [29]. Most laboratory personnel were already familiar with its use, making integration of new assays such as hepatitis B viral load testing operationally feasible. Equally, this existing infrastructure constitutes a strong platform for scale-up, thus minimizing the need for additional capital investments and human resources, leading to greater efficiencies.

Our findings further suggest that implementing HBV DNA testing on the GeneXpert platform was equally cost-comparable to centralized testing. Specifically, the unit cost per test was USD 15.37 for the GeneXpert platform, closely matching the USD 15.26 cost associated with centralized testing. This cost equivalence is partly due to efficiencies from eliminating sample transportation and logistical overheads required for the central laboratory processing. PoC testing can also substantially reduce costs associated with repeated patient visits for result collection and losses to follow-up, as results are typically available the same day, as has been documented in similar evaluations of HIV viral testing on GeneXpert PoC platforms [30]. PoC testing also improves efficiencies by eliminating the numerous steps involved in conventional viral load testing, such as packaging for dispatch, transportation from the health facility to the hub, then to the central laboratory for processing, followed by sample reception and processing, and finally, the return of results to the facility and the patient. This translates into lower personal costs for pregnant women. We are cognizant that this costing was conducted from a provider perspective during the study period and does not represent a full economic or societal cost. Indirect expenses, including essential equipment maintenance, long-term logistical support, and patient-incurred costs, were not captured, but are likely lower than under the counterfactual.

Furthermore, in our study, we observed complete concordance in sensitivity and specificity between the Xpert platform and a conventional centralized testing platform. This finding is consistent with results reported from other diverse settings, including India, The Gambia, and other regions [19,31,32]. Also, clinicians reported that the GeneXpert platform was easy to link to existing results transmission systems, further enhancing its use for clinical decision-making at sites where GeneXpert results were transmitted to clinicians via an SMS notification system that alerts clinicians when results are ready. This is consistent with findings from Tanzania, where the use of Xpert SMS enabled rapid results delivery [33].

Nevertheless, in this operational research, not all eligible pregnant women were linked to prophylaxis, nor were all infants vaccinated, especially those at greater risk of infection. Uptake of hepatitis B prophylaxis in African antenatal settings is constrained by multiple interrelated health system and demand-side barriers that extend beyond diagnostic availability. Studies from sub-Saharan Africa report that financial barriers, inadequate counselling, and limited appreciation among mothers for the risks of hepatitis infection have been shown to reduce maternal preparedness for hepatitis B prophylaxis, even when women are diagnosed during pregnancy [34]. In addition, competing service delivery priorities, particularly the stronger institutionalization of HIV and syphilis PMTCT programs, contribute to lower prioritization of HBV services within routine antenatal care [35]. Health system constraints, including gaps in trained human resources, inconsistent commodity availability, and weak documentation and follow-up mechanisms, further limit completion of the HBV PMTCT cascade [36]. Together, these findings highlight that improving diagnostic access through point-of-care testing must be accompanied by broader programmatic strengthening to achieve meaningful gains in Hepatitis B management.

This is one of the first studies providing a comprehensive field assessment of the GeneXpert platform for hepatitis B viral load testing PoC within the context of PMTCT. Although the analytical performance of the Xpert HBV Viral Load assay has been previously established, this study provides operational experience on the use of point-of-care HBV viral load testing within routine antenatal care services in resource-constrained settings. In Uganda, the GeneXpert platform is widely deployed for HIV, tuberculosis, HPV, and COVID-19 testing but has not previously been applied to HBV viral load testing in antenatal services. Despite national policies promoting integrated testing and treatment under the triple elimination strategy, access to HBV viral load testing and timely initiation of prophylaxis remain suboptimal in high-volume referral facilities. By documenting real-world implementation considerations, including workflow integration, turnaround time, care linkage, and cost implications in the absence of WHO prequalification, this study illustrates how decentralizing viral load testing may help address delays inherent in centralized systems and offers practical insights to inform context-appropriate adaptations to HBV testing algorithms, workflow integration, and programmatic planning, including supply allocation and service organization.

High-volume regional referral hospitals (RRHs) were selected as study sites because they are the main referral points for HBV prevention of mother-to-child transmission (PMTCT) services and have existing laboratory infrastructure and trained personnel for molecular diagnostic workflows. While this selection could introduce site-level performance differences aligned with resource availability, it reflects the most realistic initial implementation setting for point-of-care viral load integration in Uganda. Since many facilities with GeneXpert instruments similarly have trained laboratory personnel and midwives, the operational insights from these settings remain broadly relevant for national adoption pathways, including lower-volume sites with comparable capacity.

A key strength of this study lies in its qualitative and quantitative approach, combining laboratory performance data with insights from frontline health workers and facility teams, enabling a well-rounded understanding of the technical performance and operational feasibility. The study also incorporated unit costs per test analysis, which offer valuable inputs for budgeting and planning by program implementers and policymakers. Moreover, integrating findings related to health worker usability, turnaround time, and health system compatibility provides practical, context-specific recommendations for future adoption and scale-up.

However, several limitations should be noted. First, the sample size was modest, with only 181 pregnant women tested, which may limit the generalizability of the findings to lower-volume or more rural facilities. Second, while the unit cost

analysis captured direct testing costs, it did not account for broader cost-effectiveness dimensions such as equipment maintenance, staff time, supply chain logistics, or long-term savings from improved outcomes. The study was also limited to high-volume district and regional referral-level facilities with existing high-module GeneXpert infrastructure, which may not reflect operational realities in lower-level health centers. Furthermore, the findings should be interpreted within the broader operational constraints of delivering antenatal services in resource-limited settings, where feasibility is shaped not only by diagnostic performance but also by policy frameworks, infrastructure, supply-chain reliability, human-resource capacity, and routine documentation practices. While the study documented service-level experience with integrating hepatitis B viral load testing into routine antenatal care, gaps were observed in downstream care linkage and documentation. Of the 22 pregnant women eligible for antiviral prophylaxis, only 14 were documented to have initiated treatment, and birth records were available for only 15 women, including missing records for 7 women eligible for prophylaxis. These losses reflect the pragmatic design of the study, in which antenatal, delivery, and postnatal services followed routine standards of care without active follow-up by the study team. The incomplete initiation of prophylaxis and missing birth documentation highlight systemic weaknesses in care continuity, monitoring, and record-keeping that extend beyond the diagnostic intervention itself. Addressing these broader programmatic and administrative factors is essential to realize the full potential of point-of-care viral load testing within maternal and child health services.

## Conclusions

This study demonstrates that the GeneXpert platform is a feasible, efficient, and cost-comparable solution for decentralizing hepatitis B viral load testing within antenatal care settings. The platform substantially reduced turnaround time for results, enabling same-day clinical decision-making for the majority of women tested. With high diagnostic concordance comparable to centralized platforms, GeneXpert supports timely linkage to prophylaxis, an essential step in preventing mother-to-child transmission of hepatitis B. Leveraging a country's existing GeneXpert infrastructure, already deployed for TB, HPV, HIV, and COVID-19 testing, can offer a cost-effective pathway for integrating HBV DNA testing into routine maternal health services. However, to maximize its impact, complementary investments are required in staff capacity-building, sample prioritization protocols, and commodity security. As countries move toward the elimination of vertical transmission of hepatitis B, decentralized approaches such as PoC testing can play a catalytic role in expanding diagnostic access, accelerating treatment initiation, and supporting equitable health outcomes. Future studies should explore implementation in lower-level facilities, patient adherence to prophylaxis, and the long-term health and economic impacts of integrating PoC HBV testing into PMTCT programs.

## Supporting information

**S1 File. Clinical performance evaluation and results from Phase I.**
(DOCX)

**S2 File. Maternal Sociodemographic characteristics and other relevant background information.**
(DOCX)

**S3 File. Flow diagram showing different time points from sample collection to results receipt by pregnant women (mothers).**
(DOCX)

**S4 File. Itemised Costs for HBV VL Testing: COBAS vs GeneXpert Platforms.**
(DOCX)

**S1 Data. Data.**
(XLSX)

## Acknowledgments

We sincerely thank the facility and study teams for their dedication to patient care. We also appreciate the Ministry of Health teams from the Central Public Health Laboratories (CPHL) and the AIDS Control Program (ACP) for supporting key operational aspects of this work. The technical teams from the Clinton Health Access Initiative's Global Hepatitis, Analytics and Implementation and Research (AIR) and Global Diagnostics programs provided invaluable guidance throughout the process. We are especially grateful to Amy Azania, Caroline Boeke, Emi Okamoto, Grace Singh, Robia Islam, and Umesh Chawla for thoroughly reviewing the study protocols and sharing relevant insights from other countries that helped shape the study design. We further thank the administrative and clinical teams across the 10 participating health facilities for leading the implementation and supporting data collection. Lastly, we wish to appreciate Cepheid and their technical team for the in-kind co-financing of the study with commodities for the Xpert platform, resources for study coordination, and technical support in troubleshooting challenges with using the device for a new assay.

## Author contributions

**Conceptualization:** Linda Kisaakye Nabitaka, Viola Kasone, Emmanuel Olal, Judith Kyokushaba, Ponsiano Ocama, Isaac Ssewanyana.

**Data curation:** Viola Kasone, Emmanuel Olal, Philip Kasibante.

**Formal analysis:** Linda Kisaakye Nabitaka, Viola Kasone, Emmanuel Olal, Philip Kasibante.

**Investigation:** Linda Kisaakye Nabitaka, Viola Kasone, Emmanuel Olal, Judith Kyokushaba, Susan Nambozo.

**Methodology:** Linda Kisaakye Nabitaka, Viola Kasone, Emmanuel Olal, Judith Kyokushaba, Philip Kasibante, Ponsiano Ocama.

**Project administration:** Susan Nambozo, Aidarayaan Namakula, Isaac Ssewanyana.

**Resources:** Linda Kisaakye Nabitaka, Aidarayaan Namakula, Susan Nabadda Ndidde, Isaac Ssewanyana.

**Supervision:** Linda Kisaakye Nabitaka, Viola Kasone, Aidarayaan Namakula, Susan Nabadda Ndidde, Ponsiano Ocama, Isaac Ssewanyana.

**Validation:** Linda Kisaakye Nabitaka, Emmanuel Olal, Judith Kyokushaba, Philip Kasibante, Susan Nambozo, Aidarayaan Namakula, Susan Nabadda Ndidde, Ponsiano Ocama, Isaac Ssewanyana.

**Visualization:** Philip Kasibante.

**Writing – original draft:** Viola Kasone, Emmanuel Olal.

**Writing – review & editing:** Linda Kisaakye Nabitaka, Viola Kasone, Emmanuel Olal, Judith Kyokushaba, Philip Kasibante, Susan Nambozo, Aidarayaan Namakula, Susan Nabadda Ndidde, Ponsiano Ocama, Isaac Ssewanyana.

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
