## [Decision Letter · Decision Letter 0]

23 Dec 2025

PGPH-D-25-02257

Optimizing Hepatitis B Diagnosis for Mothers in a Low-Resource Setting: A Field Pilot of Xpert® HBV Viral Load Point-of-Care Testing in Ugandan Antenatal Clinics

Dear Dr. Kisaakye,

Thank you for submitting your manuscript to PLOS Global Public Health. After careful consideration, we feel that it has merit but does not fully meet PLOS Global Public Health’s publication criteria as it currently stands. Therefore, we invite you to submit a revised version of the manuscript that addresses the points raised during the review process.

Cohen's Kappa (κ) statistics can be used for inter-assay reliability. The study should also highlight turnaround time differences between the assays, if any, impacting actual patient care and outcomes

We look forward to receiving your revised manuscript.

Kind regards,

Mustafa A Barbhuiya, PhD

Guest Editor

Journal Requirements:

Additional Editor Comments (if provided):

Cohen's Kappa (κ) statistics can be used for inter-assay reliability.

The study should also highlight turnaround time differences between the assays, if any, impacting actual patient care and outcomes

Reviewers' comments:

Reviewer's Responses to Questions

**Comments to the Author**

1. Does this manuscript meet PLOS Global Public Health’s publication criteria? Is the manuscript technically sound, and do the data support the conclusions? The manuscript must describe methodologically and ethically rigorous research with conclusions that are appropriately drawn based on the data presented.? Is the manuscript technically sound, and do the data support the conclusions? The manuscript must describe methodologically and ethically rigorous research with conclusions that are appropriately drawn based on the data presented.

Reviewer #1: Yes

Reviewer #2: Partly

Reviewer #3: Yes

2. Has the statistical analysis been performed appropriately and rigorously?

Reviewer #1: Yes

Reviewer #2: I don't know

Reviewer #3: I don't know

3. Have the authors made all data underlying the findings in their manuscript fully available (please refer to the Data Availability Statement at the start of the manuscript PDF file)?

The PLOS Data policy requires authors to make all data underlying the findings described in their manuscript fully available without restriction, with rare exception. The data should be provided as part of the manuscript or its supporting information, or deposited to a public repository. For example, in addition to summary statistics, the data points behind means, medians and variance measures should be available. If there are restrictions on publicly sharing data—e.g. participant privacy or use of data from a third party—those must be specified.requires authors to make all data underlying the findings described in their manuscript fully available without restriction, with rare exception. The data should be provided as part of the manuscript or its supporting information, or deposited to a public repository. For example, in addition to summary statistics, the data points behind means, medians and variance measures should be available. If there are restrictions on publicly sharing data—e.g. participant privacy or use of data from a third party—those must be specified.

Reviewer #1: Yes

Reviewer #2: Yes

Reviewer #3: Yes

4. Is the manuscript presented in an intelligible fashion and written in standard English?

Reviewer #1: Yes

Reviewer #2: Yes

Reviewer #3: Yes

5. Review Comments to the Author

Reviewer #1: 1. The manuscript should more prominently highlight the substantial loss to prophylaxis follow-up (only 14 of the 22 pregnant women eligible for antiviral prophylaxis received it) and the absence of birth-record follow-up (birth records were available for only 15 of the 181 pregnant women tested). It is important to address the reasons for these gaps, as they significantly limit the ability to draw meaningful conclusions about the intervention’s effectiveness in improving overall maternal–infant care linkage. Additionally, please correct the statement in line 324: “we found that 15 of the 22 pregnant women eligible for prophylaxis had received it.” Based on the previously reported data, the number of women who received prophylaxis was 14.

2. The study was co-financed by Cepheid, the manufacturer of the Xpert® assay, which also supplied the test cartridges. It would be best practice to expand the discussion to clarify how potential conflicts of interest were managed and how objectivity was ensured. Furthermore, does the reported cost per test (USD 15.37) account for indirect expenses such as essential equipment maintenance, long-term logistical support, and patient-incurred costs? While cartridges for this study were provided by the manufacturer, it would be useful to discuss the long-term sustainability and cost implications beyond the study period.

Reviewer #2: The authors have conducted an extensive assessment of the functional feasibility of the Xpert® HBV Viral Load (VL) point-of-care assay in comparison with the reference COBAS® AmpliPrep/COBAS TaqMan® platform. However, the study lacks substantive scientific novelty. Both diagnostic kits have already undergone comprehensive validation and regulatory evaluation prior to market approval, and their analytical performance, accuracy, and intended use are clearly described in their respective manufacturer manuals.

In this context, evaluating the operational feasibility of an already-established and commercially validated diagnostic platform adds limited new scientific value. Operational feasibility is largely determined by broader programmatic and administrative factors—such as government policy, infrastructure availability, supply-chain efficiency, human-resource capacity, and national implementation strategy—rather than by laboratory-based comparative assessments alone. Without integrating these real-world determinants, the study’s contribution remains narrow.

The work would have offered greater novelty if the authors had developed:

1) Modified or optimized methods using these existing kits tailored for resource-constrained or operationally challenging hospital settings; or

2) Cost-effective alternative workflows or algorithms that improve accessibility or turnaround time beyond standard manufacturer recommendations.

Another potential area of novelty could have been evaluating clinical impact, for example assessing whether diagnosing pregnant women with these assays leads to measurable improvements in HBV prophylaxis outcomes in newborns. Such analyses would extend the study beyond analytical comparison toward clinically meaningful and policy-relevant insights.

Reviewer #3: Dear Authors,

Thank you for submitting your manuscript. I found your study to be relevant, given the ongoing challenges in scaling up hepatitis B diagnosis and prevention in low-resource settings. The findings on decentralized testing with GeneXpert are timely and could inform practice in Uganda and beyond.

To further strengthen your manuscript, I suggest clarifying the rationale for selecting high-volume hospitals and discussing how this might affect generalizability. It would also be helpful to expand on the diagnostic accuracy of the Xpert® assay, including the limit of detection and any discrepant results. More detail on the process for obtaining informed consent, as well as reasons for non-uptake of prophylaxis, would add depth. Discussing barriers to full uptake of prophylaxis and newborn vaccination, with references to local or regional studies, would strengthen the context. Finally, addressing the generalizability of your findings to other low-resource settings, and acknowledging limitations like sample size and selection bias, would help readers better interpret your results.

With these improvements, your paper will be a valuable contribution to the literature on hep B prevention and point-of-care diagnostics in resource-limited settings.

6. PLOS authors have the option to publish the peer review history of their article (what does this mean?). If published, this will include your full peer review and any attached files.). If published, this will include your full peer review and any attached files.

**Do you want your identity to be public for this peer review?** For information about this choice, including consent withdrawal, please see our Privacy Policy..

Reviewer #1: No

Reviewer #2: No

Reviewer #3: No

Figure Resubmissions:

---

## [Decision Letter · Decision Letter 1]

11 Mar 2026

PGPH-D-25-02257R1

Optimizing Hepatitis B Diagnosis for Mothers in a Low-Resource Setting: A Field Pilot of Xpert® HBV Viral Load Point-of-Care Testing in Ugandan Antenatal Clinics

Dear Dr. Kisaakye,

Thank you for submitting your manuscript to PLOS Global Public Health. After careful consideration, we feel that it has merit but does not fully meet PLOS Global Public Health’s publication criteria as it currently stands. Therefore, we invite you to submit a revised version of the manuscript that addresses the points raised during the review process.

Please address the minor comments raised by Reviewer 4.

We look forward to receiving your revised manuscript.

Kind regards,

Emma Campbell, Ph.D

Staff Editor

On behalf of

Mustafa A Barbhuiya, PhD

Academic Editor

Journal Requirements:

1. Please provide a detailed online Financial Disclosure statement. This is published with the article. It must therefore be completed in full sentences and contain the exact wording you wish to be published.

a) Please clarify all sources of financial support for your study. List the grants, grant numbers, and organizations that funded your study, including funding received from your institution. Please note that suppliers of material support, including research materials, should be recognized in the Acknowledgements section rather than in the Financial Disclosure.

b) State the initials, alongside each funding source, of each author to receive each grant. For example: “This work was supported by the National Institutes of Health (####### to AM; ###### to CJ) and the National Science Foundation (###### to AM).”

c) State what role the funders took in the study. If the funders had no role in your study, please state: “The funders had no role in study design, data collection and analysis, decision to publish, or preparation of the manuscript.”

For more information, please go to our submission guidelines:

https://journals.plos.org/climate/s/submission-guidelines#loc-financial-disclosure-statement

For more information, please go to our submission guidelines:

https://journals.plos.org/globalpublichealth/s/submission-guidelines#loc-financial-disclosure-statement

2. Please ensure that the funders and grant numbers match between the Financial Disclosure field and the Funding Information tab in your submission form. Note that the funders must be provided in the same order in both places as well.

3. Please update your online Competing Interests statement. If you have no competing interests to declare, please state: "The authors have declared that no competing interests exist."

4. We note that your Data Availability Statement is currently as follows: "All data has been uploaded as supplementary files."

Please confirm at this time whether or not your submission contains all raw data required to replicate the results of your study. Authors must share the “minimal data set” for their submission. PLOS defines the minimal data set to consist of the data required to replicate all study findings reported in the article, as well as related metadata and methods (https://journals.plos.org/globalpublichealth/s/data-availability#loc-minimal-data-set-definition).

If your submission does not contain these data, please either upload them as Supporting Information files or deposit them to a stable, public repository and provide us with the relevant URLs, DOIs, or accession numbers. For a list of recommended repositories, please see https://journals.plos.org/globalpublichealth/s/recommended-repositories.

5. Please ensure that the Title in your manuscript and the Title in your online submission form are the same.

6. Please provide separate main figure files in .tif or .eps format only and ensure that all files are under our size limit of 10MB.

7. We have noticed that you have uploaded Supporting Information files, but you have not included a list of legends. Please add a full list of legends for your Supporting Information files before or after the references list.

8. Some material included in your submission may be copyrighted. According to PLOS’s copyright policy, authors who use figures or other material (e.g., graphics, clipart, maps) from another author or copyright holder must demonstrate or obtain permission to publish this material under the Creative Commons Attribution 4.0 International (CC BY 4.0) License used by PLOS journals. Please closely review the details of PLOS’s copyright requirements here: PLOS Licenses and Copyright. If you need to request permissions from a copyright holder, you may use PLOS's Copyright Content Permission form.

Potential Copyright Issues:

Figure 1: Please confirm whether you drew the images / clip-art within the figure panels by hand. If you did not draw the images, please provide (a) a link to the source of the images or icons and their license / terms of use; or (b) written permission from the copyright holder to publish the images or icons under our CC-BY 4.0 license. Alternatively, you may replace the images with open source alternatives. See these open source resources you may use to replace images / clip-art:

- https://openclipart.org/

We do not publish any copyright or trademark symbols that usually accompany proprietary names, eg (R), (C), or TM  (e.g. next to drug or reagent names). Please remove all instances of trademark/copyright symbols throughout the text, including © on page 30.

Additional Editor Comments (if provided):

Please address the minor comments from the reviewer 4.

Reviewers' comments:

Reviewer's Responses to Questions

**Comments to the Author**

1. If the authors have adequately addressed your comments raised in a previous round of review and you feel that this manuscript is now acceptable for publication, you may indicate that here to bypass the “Comments to the Author” section, enter your conflict of interest statement in the “Confidential to Editor” section, and submit your "Accept" recommendation.

Reviewer #1: All comments have been addressed

Reviewer #2: All comments have been addressed

Reviewer #4: (No Response)

2. Does this manuscript meet PLOS Global Public Health’s publication criteria? Is the manuscript technically sound, and do the data support the conclusions? The manuscript must describe methodologically and ethically rigorous research with conclusions that are appropriately drawn based on the data presented.? Is the manuscript technically sound, and do the data support the conclusions? The manuscript must describe methodologically and ethically rigorous research with conclusions that are appropriately drawn based on the data presented.

Reviewer #1: Yes

Reviewer #2: Yes

Reviewer #4: Partly

3. Has the statistical analysis been performed appropriately and rigorously?

Reviewer #1: Yes

Reviewer #2: Yes

Reviewer #4: Yes

4. Have the authors made all data underlying the findings in their manuscript fully available (please refer to the Data Availability Statement at the start of the manuscript PDF file)?

The PLOS Data policy requires authors to make all data underlying the findings described in their manuscript fully available without restriction, with rare exception. The data should be provided as part of the manuscript or its supporting information, or deposited to a public repository. For example, in addition to summary statistics, the data points behind means, medians and variance measures should be available. If there are restrictions on publicly sharing data—e.g. participant privacy or use of data from a third party—those must be specified.requires authors to make all data underlying the findings described in their manuscript fully available without restriction, with rare exception. The data should be provided as part of the manuscript or its supporting information, or deposited to a public repository. For example, in addition to summary statistics, the data points behind means, medians and variance measures should be available. If there are restrictions on publicly sharing data—e.g. participant privacy or use of data from a third party—those must be specified.

Reviewer #1: Yes

Reviewer #2: Yes

Reviewer #4: Yes

5. Is the manuscript presented in an intelligible fashion and written in standard English?

Reviewer #1: Yes

Reviewer #2: Yes

Reviewer #4: Yes

6. Review Comments to the Author

Reviewer #1: Thank you for the revisions. All prior comments have been adequately addressed.

Reviewer #2: The author has significantly improved and presented the results in intelligible fashion. I recommend this paper for publication

Reviewer #4: General comments:

This manuscript describes a demonstration project of using a hepatitis B point-of-care (POC) test in Ugandan antenatal clinics (ANCs).

Quantitatively, I felt this manuscript was light, though this should not necessarily devalue the contribution of this manuscript. Nevertheless, I have a few quantitatively-minded questions.

I noticed that there was no mention that the samples for phase I were randomly selected (lines 117-121). I am somewhat worried that there may be a bias if these samples were not randomly selected. Also, these samples appear to have been banked and are not necessarily like samples from a POC test. Can you speak to whether there could be any bias from using archived samples versus "fresh" samples?

I also felt as though some information was lacking regarding the turnaround time. Do you have more specific information on the turnaround time (lines 217-221)? It's good that many were returned same day, but I would have liked to have known how quickly the whole process took. That would seem to be extremely helpful for others trying to implement this process. In addition, why were those not able to be returned same day? It seems like the table in line 223 could be displayed as a flow chart where we could see where the samples failed in the process to be returned on the same day.

Finally, I thought the qualitative results could be more quantitative. For example, take the paragraph on lines 248-251. The phrases "Most sites", "a few facilities", and "Most facilities" could be converted from vague statements into quantitative results. I encourage you to read through this second and be slightly more quantitative with your presentation.

Specific comments:

1. (lines 52-53) I don't think this is correct. The UNAIDS factsheet (https://www.unaids.org/en/resources/fact-sheet) says 1.3 million people acquired HIV in 2024. Divided by 365, that is approximately 3,562 people per day. The same source says new infections among children were 120,000, which is approximately 329 HIV infections per day. That's much closer, but still almost 2.5 times more infections.

2. (lines 169-170) Please provide methodological citations for Deming regression and Bland-Altman plots since some readers may not be familiar with these methods.

3. (lines 226-227) I recommend reporting these results in ascending order by viral load.

7. PLOS authors have the option to publish the peer review history of their article (what does this mean?). If published, this will include your full peer review and any attached files.). If published, this will include your full peer review and any attached files.

**Do you want your identity to be public for this peer review?** For information about this choice, including consent withdrawal, please see our Privacy Policy..

Reviewer #1: No

Reviewer #2: No

Reviewer #4: No

 Figure Resubmissions:

---

## [Editor Report · Decision Letter 2]

13 Apr 2026

Optimizing Hepatitis B Diagnosis for Mothers in a Low-Resource Setting: A Field Pilot of Xpert Point-of-Care Viral Load Testing in Ugandan Antenatal Clinics

PGPH-D-25-02257R2

Dear Dr. Kisaakye,

We are pleased to inform you that your manuscript 'Optimizing Hepatitis B Diagnosis for Mothers in a Low-Resource Setting: A Field Pilot of Xpert Point-of-Care Viral Load Testing in Ugandan Antenatal Clinics' has been provisionally accepted for publication in PLOS Global Public Health.

Best regards,

Mustafa A Barbhuiya, PhD

Academic Editor